# Distal Nerve Transfers in High Peroneal Nerve Lesions: An Anatomical Feasibility Study

**DOI:** 10.3390/jpm13020344

**Published:** 2023-02-16

**Authors:** Stefan Salminger, Clemens Gstoettner, Lena Hirtler, Roland Blumer, Christoph Fuchssteiner, Gregor Laengle, Johannes A. Mayer, Konstantin D. Bergmeister, Wolfgang J. Weninger, Oskar C. Aszmann

**Affiliations:** 1AUVA Trauma Hospital Lorenz Böhler—European Hand Trauma Center, Donaueschingenstrasse 13, 1200 Vienna, Austria; 2Deparment of Plastic and Reconstructive Surgery, Clinical Laboratory for Bionic Extremity Reconstruction, Medical University of Vienna, 1090 Vienna, Austria; 3Department of Plastic and Reconstructive Surgery, Medical University of Vienna, 1090 Vienna, Austria; 4Center for Anatomy and Cell Biology, Division of Anatomy, Medical University of Vienna, 1090 Vienna, Austria; 5Department of Plastic, Aesthetic and Reconstructive Surgery, Karl Landsteiner University of Health Sciences, University Hospital St. Poelten, 3100 Krems, Austria

**Keywords:** peroneal nerve lesion, drop foot, nerve transfer, axon count

## Abstract

The peroneal nerve is one of the most commonly injured nerves of the lower extremity. Nerve grafting has been shown to result in poor functional outcomes. The aim of this study was to evaluate and compare anatomical feasibility as well as axon count of the tibial nerve motor branches and the tibialis anterior motor branch for a direct nerve transfer to reconstruct ankle dorsiflexion. In an anatomical study on 26 human body donors (52 extremities) the muscular branches to the lateral (GCL) and the medial head (GCM) of the gastrocnemius muscle, the soleus muscle (S) as well as the tibialis anterior muscle (TA) were dissected, and each nerve’s external diameter was measured. Nerve transfers from each of the three donor nerves (GCL, GCM, S) to the recipient nerve (TA) were performed and the distance between the achievable coaptation site and anatomic landmarks was measured. Additionally, nerve samples were taken from eight extremities, and antibody as well immunofluorescence staining were performed, primarily evaluating axon count. The average diameter of the nerve branches to the GCL was 1.49 ± 0.37, to GCM 1.5 ± 0.32, to S 1.94 ± 0.37 and to TA 1.97 ± 0.32 mm, respectively. The distance from the coaptation site to the TA muscle was 43.75 ± 12.1 using the branch to the GCL, 48.31 ± 11.32 for GCM, and 19.12 ± 11.68 mm for S, respectively. The axon count for TA was 1597.14 ± 325.94, while the donor nerves showed 297.5 ± 106.82 (GCL), 418.5 ± 62.44 (GCM), and 1101.86 ± 135.92 (S). Diameter and axon count were significantly higher for S compared to GCL as well as GCM, while regeneration distance was significantly lower. The soleus muscle branch exhibited the most appropriate axon count and nerve diameter in our study, while also reaching closest to the tibialis anterior muscle. These results indicate the soleus nerve transfer to be the favorable option for the reconstruction of ankle dorsiflexion, in comparison to the gastrocnemius muscle branches. This surgical approach can be used to achieve a biomechanically appropriate reconstruction, in contrast to tendon transfers which generally only achieve weak active dorsiflexion.

## 1. Introduction

The peroneal nerve is one of the most commonly injured nerves of the lower extremity [1]. In traumatic cases, often resulting from knee luxation, the stretch lesion of the nerve can lead to the loss of intraneural architecture reaching from the distal thigh to the fibular neck [2,3]. Injuries to the deep peroneal nerve result in paralysis of the muscles in the anterior compartment with associated loss of toe and ankle dorsiflexion as well as loss of sensation in the dorsal first web space of the foot [4]. The resulting “drop foot” leads to significant gait disturbances, necessitating a high stepping gait in order to lift the foot fully from the ground. Tendon transfers and ankle foot orthotics have been used widely as primary and secondary measures for the treatment of drop foot [4,5,6]. Both methods primarily aim to stabilize the ankle, preventing the foot from falling and thus facilitating an improved gait cycle. However, both fail to provide adequate ankle dorsiflexion. The most commonly performed tendon transfer uses the tibialis posterior muscle as a motor for ankle dorsiflexion. Even if the patient regains active dorsiflexion of the foot, it will remain weak and therefore not completely correct the gait problem [7].

Standard nerve repair consists of the reconstruction of the injured nerve with autologous nerve grafts. However, nerve grafting of peroneal nerve injuries has led to poor functional outcomes, in particular when grafts longer than six centimeters were used [8,9]. Some authors have proposed performing tibialis posterior tendon transfer additionally to a traditional nerve repair, to increase the likelihood of functional ankle dorsiflexion [10].

To overcome the frustration of standard nerve grafting in the reconstruction of peroneal nerve lesions, distal nerve transfers using expendable donor nerves in the lower limb have become increasingly popular [2,4,11,12,13,14]. Due to the shorter distance of regeneration from donor axons to the targeted motor end plates and the use of pure motor donors, distal nerve transfers facilitate faster recovery and may also reduce the amount of surgical dissection [15]. The use of a partial tibial nerve transfer in patients suffering from extensive peroneal nerve lesions has been described in anatomical studies and small case series [2,4,11,12,13,14]. However, there is limited clinical and anatomical experience regarding which branch of the tibial nerve qualifies best for this nerve transfer, considering properties such as axon count and distance to the motor entry point (MEP) of the tibialis anterior muscle [4,16]. Mackinnon et al., demonstrated the importance of matching appropriately sized nerves and axon counts in the reconstruction of the peroneal nerve [17]. Thus, the aim of this study was to evaluate and compare anatomical feasibility as well as axon count match between the tibial nerve motor branches (donor) and the tibialis anterior motor branch (recipient) for a direct nerve transfer. This study should therefore determine the best suitable option for distal nerve transfer in high peroneal nerve lesions.

## 2. Material and Methods

The study was conducted at the Division of Anatomy of the Medical University of Vienna and was approved by the local institutional review board. A total of 52 individual fresh lower extremities of 26 human body donors (11 females, 15 males) were included. The mean age of the donors was 80.85 ± 7.67 years (range 69–97 years). The only exclusion criterium was defined as no prior surgery with a visible scar within the popliteal region. All of these individuals had donated their bodies to medical education and research at the Center of Anatomy and Cell Biology, Medical University of Vienna.

### 2.1. Anatomical Dissection and Nerve Transfers

The skin incision was made in a z-wise manner from the distal posterior thigh over the popliteal fossa until below the fibular neck. Using standard surgical instruments, the tibial nerve and its branches to the muscles of the lower leg as well as the common peroneal nerve and its bifurcation into superficial and deep peroneal nerves were dissected. The single muscular branches to the lateral and the medial head of the gastrocnemius muscle, the soleus muscle as well as the tibialis anterior muscle were identified and each nerve’s external diameter was measured using a digital caliper with a measuring accuracy of 1/100 (0.01) millimeter. The femorotibial joint was then cannulated to determine its exact plane. The distance from the femorotibial joint to the MEP of the individual muscle branches was measured. The nerve branches to the medial gastrocnemius, lateral gastrocnemius, and soleus muscles were subsequently cut at the level of the epimysium. Then, the nerve transfers of each individual donor to the tibialis anterior motor branch were performed. Where necessary, the tibialis anterior branch was dissected proximally from the common peroneal nerve, in order to achieve tensionless coaptation without the need for a graft. For each nerve transfer, it was documented whether the achievable coaptation site reached the fibular head, the peroneal bifurcation, and the tibialis anterior MEP, respectively. Furthermore, the distance between the coaptation point of the transfer to the MEP of the tibialis anterior muscle was measured (Figure 1a,b).

### 2.2. Tissue Collection

In eight out of the 52 lower extremities from four consecutive fresh body donors, the four individual muscle branches (GCM, GCL, S, TA) were harvested for axon count. The mean age of the five donors was 87.6 ± 9.18 years ranging from 72 to 97 years. In total, 32 nerve specimens were analyzed. 

### 2.3. Tissue Preparation

Nerves were immersion-fixed for 24 h at 4 °C in 4% paraformaldehyde in 0.1 M phosphate buffer at pH 7.4. Thereafter, tissue was rinsed in phosphate-buffered saline (PBS) at pH 7.4, cryo-embedded, and kept at −80 °C before further processing. A detailed description of cryo-embedding of tissue is provided by Blumer et al. [18]. Cryo-embedded tissue was used for immunofluorescence and analyzed in the confocal laser scanning microscope (CLSM).

### 2.4. Antibodies

The following antibodies were used. The supplier, RRID number, and antibody concentration are provided. Chicken anti-neurofilament [NF, (Merck/Millipore, Burlington, Massachusetts, USA, RRID: AB_177520; 1:2000)], goat anti-choline acetyltransferase [ChAT, (Merck/Millipore, Burlington, MA, USA, RRID: AB_2079751, 1:100)], rabbit anti-myelin basic protein [MBP, (Merck/Millipore, Burlington, MA, USA, RRID: AB_94975, 1:100)] and rabbit ant tyrosine hydroxylase [TH, (Merck/Millipore, Burlington, MA, USA, RRID: AB_390204, 1:250)].

### 2.5. Immunofluorescence

We performed one single labeling experiment incubating with anti-NF and three double labeling experiments incubating with anti-NF in combination with anti-MBP, anti-ChAT, and anti-TH. Before immunolabeling, frozen cross sections at 10 µm thickness were cut with a Cryocut (Leica CM1950; Leica Microsystems GmbH, Wetzlar, Germany) followed by drying at room temperature for 30 min. Then, tissue was blocked for 2 hours in 10% normal goat serum (single staining with anti-NF and double stainings with anti-NF/anti-MBP and anti-NF/anti-TH) or normal rabbit serum (double staining with anti-NF/anti-ChAT) in PBS containing 0.1% Triton (PBS-T). Thereafter, sections were incubated for 48 hours with the primary antibodies at 4 °C, washed with PBS-T, and incubated for 4 hours with the AlexaFluor 488 and 568 conjugated secondary antibodies at concentrations of 1:500 at 37 °C. Finally, the tissue was rinsed again in PBS-T and mounted in a fluorescence mounting medium. 

Fluorescently labeled sections were analyzed with a confocal laser scanning microscope [CLSM (Olympus FV3000, Olympus Europa SE & Co. KG, Hamburg, Germany)]. For CLSM analyses, a series of virtual sections at 0.9 µm thickness were cut through the structures of interest. Each section was photo-documented with a 1024 × 1024 pixel resolution and 3D projections were rendered using Image J software(Version 1.53t, Wayne Rasband). Single-colored images were generated using a laser with an excitation wavelength of 568 nm and double-colored images using lasers with excitation wavelengths of 488 and 568 nm. Additionally, bright-filed images were recorded in the CLSM using a transmitted light detector to show morphological details.

For negative controls, primary antibodies were omitted and secondary antibodies were used alone. In all cases, the omission of the primary antibody resulted in a complete lack of immunostaining.

### 2.6. Quantification of Nerve Fibers

For nerve fiber quantification, single-colored images were captured with the CLSM at 20× magnification. All myelinated nerve fibers were counted by using the imaging software ImageJ.

### 2.7. Statistical Analysis

Data are presented as mean ± standard deviation. All statistical analyses were performed using the Analysis ToolPak for Microsoft Excel (Microsoft office 2018, Redmond, WA, USA). Quantitative data for different nerve measurements were compared using paired *t*-tests, after visual assessment for normal distribution. Statistical significance was defined as *p* < 0.05.

## 3. Results

The individual motor branches of the tibial and peroneal nerves could be identified in all 52 lower extremities. The average nerve diameter of TA was 1.97 ± 0.32 mm. The diameter of S was 1.94 ± 0.37 mm, being significantly higher than GCL (1.49 ± 0.37 mm, *p* < 0.001) as well as GCM (1.5 ± 0.32 mm, *p* < 0.001). The average distance from the femorotibial joint to the MEP of the possible nerve donors was 38.3 ± 10.43 for the GCL, 34.7 ± 10.61 for the GCM, and 66.28 ± 10.68 millimeters for S (Table 1). When performing the individual nerve transfers, the fibular neck could be reached in 51 out of 52 extremities when using the soleus motor branch as a donor. Using the branches to the GCL and the GCM, this was possible in 30 out of 52 and 18 out of 52 extremities, respectively. The distance of nerve regeneration after nerve transfer (distance between possible coaptation site and MEP of tibialis anterior muscle) was shortest for S with 19.12 ± 11.68 mm, compared to GCL (43.75 ± 12.1 mm, *p* < 0.001) and GCM (48.31 ± 11.32 *p* < 0.001). In only three out of 52 (5.77%) extremities, the soleus branch was not able to reach the bifurcation of the deep and superficial peroneal nerve, while the GCL branch failed to reach this bifurcation in 33 (63.46%), and the GCM branch in 37 out of 52 (71.15%) extremities. Additionally, using the branch innervating the soleus muscle, in seven out of 52 extremities, the transfer reached the MEP of the TA muscle, while this was not the case for any transfers of the gastrocnemius heads (Table 2).

### 3.1. Molecular Characterization of Axons

To visualize axons, we used anti-NF. Overview images of nerve cross-sections confirmed the size differences between the putative donor nerves and the recipient nerve (Figure 2A–D). Each muscle nerve contained axons of variable thickness and thick and thin axons could be distinguished (Figure 3A–A″,B–B″,C–C″). To determine the molecular quality of axons, we performed a series of double-labeling experiments. By labeling with anti-NF and a marker for myelin (anti-MBP), we showed that thick axons were surrounded by a myelin sheath although thinner axons with myelin were also present (Figure 3A–A″). Additionally, we observed extremely thin axons (1 µm) lacking myelin. Unmyelinated axons were tightly packed together and arranged in groups (Figure 3A–A″). Double labeling with anti-NF and a marker for cholinergic axons (anti-ChAT) revealed that myelinated nerve fibers of large diameter were ChAT-positive whereas other large-diameter myelinated axons lacked ChAT immunoreactivity. There were also a few thin myelinated nerves that expressed ChAT. All unmyelinated nerve fibers lacked ChAT immunoreactivity (Figure 3B–B″). Double labeling with anti-NF and a marker for sympathetic axons (anti-TH) showed that thick myelinated axons lacked TH whereas the vast majority of unmyelinated axons expressed TH (Figure 3C–C″).

Based on the size and molecular profile, we conclude that large myelinated axons with ChAT immunoreactivity are α-motoneurons innervating skeletal muscle fibers and thinly myelinated axons expressing ChAT are γ-motoneurons innervating intrafusal muscle fibers of muscle spindles. Large myelinated axons without ChAT-immunoreactivity are sensory and most likely proprioceptive axons innervating muscle spindles and Golgi tendon organs. Unmyelinated TH-positive axons are peripheral sympathetic axons.

### 3.2. Quantification of Axons

Exclusively myelinated nerve fibers of large diameter were counted including α-motoneurons and sensory, proprioceptive axons. It was not possible to limit counts to α-motoneurons, because ChAT signals were too weak in some donor nerves due to the postmortem interval (interval between the time of death of individuals and tissue fixation). This is in accordance with a recent study where we have demonstrated that the ChAT signal is stable for 24 h after death and then decreases [19]. The TA branch showed 1597.14 ± 325.94 myelinated fibers (Table 1). In comparison, the donor nerves showed axon counts of 1101.86 ± 135.92 for S, which was significantly higher than the axon counts of both GCL (297.5 ± 106.82, *p* < 0.001) and GCM (418.5 ± 62.44, *p* < 0.001).

## 4. Discussion

The presented study clearly determines the nerve branch of the soleus muscle as the best option for reanimation of dorsiflexion in high peroneal nerve lesions. Since the tibial nerve is often spared during knee dislocation injuries, while the peroneal nerve is commonly severely damaged, the motor branches of the tibial nerve can be considered reasonable donors for reanimation of the tibialis anterior muscle in such cases [13,14,15]. The tibialis anterior and extensor hallucis longus muscles are most important for dorsiflexion of the foot and, thus, a normal gait pattern, which is why selective reinnervation of these muscles is the primary goal of reconstruction [15]. Different authors have described a partial tibial nerve transfer for functional reconstruction of ankle dorsiflexion in long-distance peroneal nerve lesions [2,4,11,12,13,14,15]. However, this is the first study to determine the nerve branch to the soleus muscle as the best option for reanimation, both from the time of reinnervation and, most importantly, regarding axon count match. The impact of replacing like with like, especially in peroneal nerve reconstruction, has been shown by Mackinnon et al. [17]. The results of this study have shown, that the nerve branch to the soleus muscle provides the best match to the tibialis anterior muscle branch, both from its nerve diameter and axon count. In comparison to the soleus branch, each of the branches to the gastrocnemius heads shows less than half of the myelinated fibers, which would lead to an increased nerve fiber mismatch (exceeding 1 to 5) between donor and recipient.

A common problem of the different donor nerve options from the tibial nerve is the counterintuitive nature of associated motions after reinnervation. However, different tendon and nerve transfers performed in the upper limb have shown excellent outcomes with little reeducation required [2]. If reeducation seems difficult, sEMG (surface electromyography) feedback can aid in reformatting the motor matrix even in difficult cases [20]. Additionally, central plasticity has been proven to enable patients to regain ankle dorsiflexion by tendon transfer of an originally antagonistic muscle such as the tibialis posterior muscle [21].

Overall, a favorable axon count ratio and short regeneration distance will allow reliable and rapid reinnervation after soleus to tibialis anterior nerve transfer. However, a disadvantage of using the soleus muscle branch is the inevitable functional deficit of nerve harvest. Nonetheless, as both heads of the gastrocnemius muscle retain their innervation and thus their function, the loss of soleus muscle function remains a reasonable sacrifice for restoration of ankle dorsiflexion and may ameliorate the force disbalance of the ankle joint. Especially when considering the alternative of tendon transfers or the use of ankle orthotics, both limiting active and passive plantarflexion in the ankle. Thus, the results of this anatomical study should introduce the soleus muscle branch transfer into the clinical routine in surgical reconstruction of high peroneal nerve lesions.

Although transfers from the branches to the flexor hallucis longus and flexor digitorum longus muscles have been reported as advantageous due to the more distal coaptation site, the results of our study have shown that the muscle branch to the soleus muscle is also able to treat very distal lesions while further providing an appropriate axon count and limiting surgical dissection [2]. In over 90% of dissected extremities, the soleus muscle branch was able to reach far distal to the bifurcation of the deep and superficial peroneal nerve, allowing a tension-free coaptation on average 19 mm proximal of the tibialis anterior MEP. Furthermore, the axon count for the flexor hallucis longus branch has been reported to be less than 50% of the tibialis anterior axon count, limiting the potential for motor recovery [12].

If a lesion of the deep peroneal nerve cannot be bypassed by the use of the soleus branch transfer, a short interpositional nerve graft could be considered. This would, however, require a disadvantageous second suture line associated with poorer results of reinnervation. In such cases, the use of a more distal but smaller muscle branch of flexor hallucis longus or flexor digitorum longus may be feasible [2]. Nonetheless, as shown by the results of this study, the soleus branch is able to reach to level distal to the fibular neck in the vast majority of extremities. Thus, it should be able to surpass the nerve damage even in extensive peroneal nerve injuries. However, in cases with sciatic nerve damage and associated axon loss within the tibial nerve, a transfer of the soleus branch would be of uncertain benefit.

In the present study, molecular analyses distinguished different nerve fiber populations in donor and recipient nerves. This includes myelinated versus unmyelinated axons as well as putative sensory, motor, and sympathetic nerve fibers. Unfortunately, immunohistochemical detection was not possible in each case, most likely because the time interval between death and tissue harvesting was too long, which impeded immunoreactivity. Nevertheless, molecular parameters of donor and recipient nerves could provide qualitative parameters to select the most suitable donor for nerve transfer to meet conditions that optimize muscle recovery following nerve injury.

## 5. Conclusions

The comparatively small difference in axon counts between donor and recipient nerves as well as its anatomical suitability reveals the soleus nerve transfer to be the best of the evaluated options for functional the reconstruction of ankle dorsiflexion in extensive peroneal nerve lesions. This surgery can be expected to provide superior results compared to the generally weak active dorsiflexion in patients after tendon transfers [7]. However, in cases of failure of reinnervation within a reasonable time frame, performing a tibialis posterior tendon transfer remains an option to restore some amount of ankle dorsiflexion [4].

## Figures and Tables

**Figure 1 jpm-13-00344-f001:**
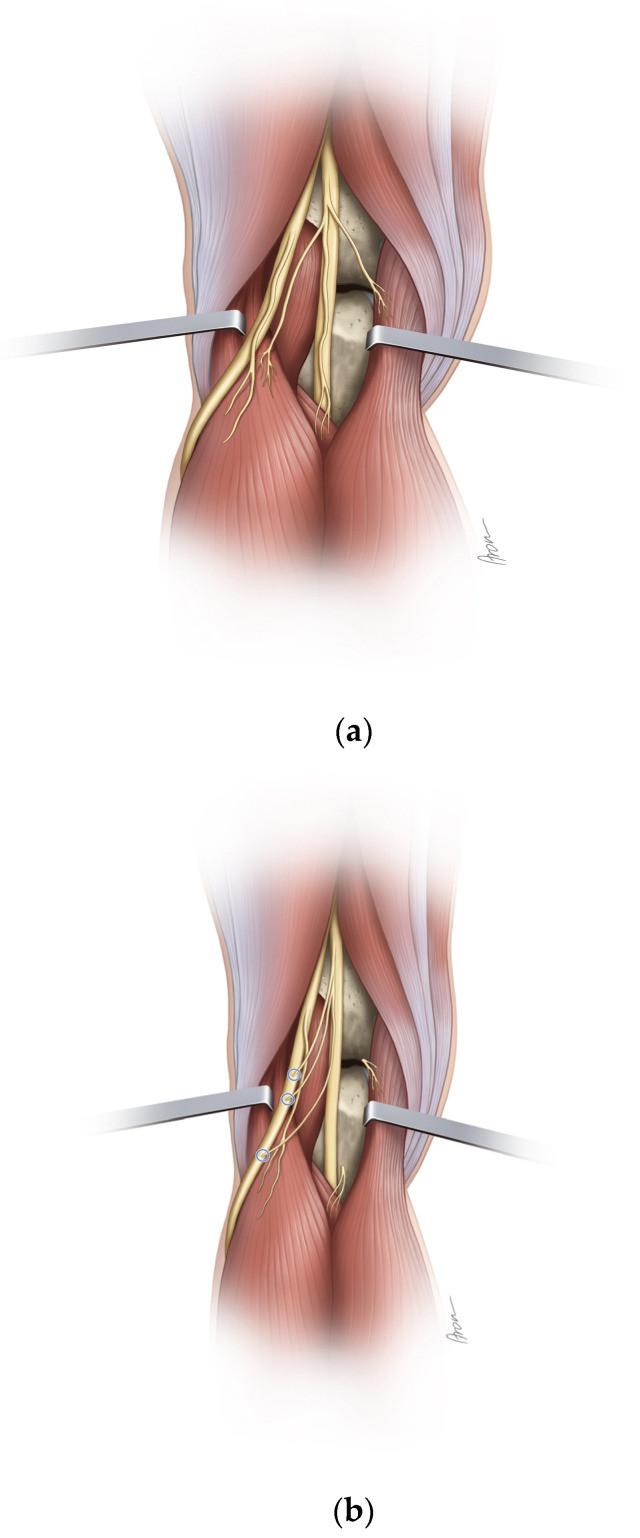
(**a**,**b**) Schematical illustration of the different nerve transfers in correlation to the femorotibial joint as well as the fibular head before (**a**) and after (**b**) transfer.

**Figure 2 jpm-13-00344-f002:**
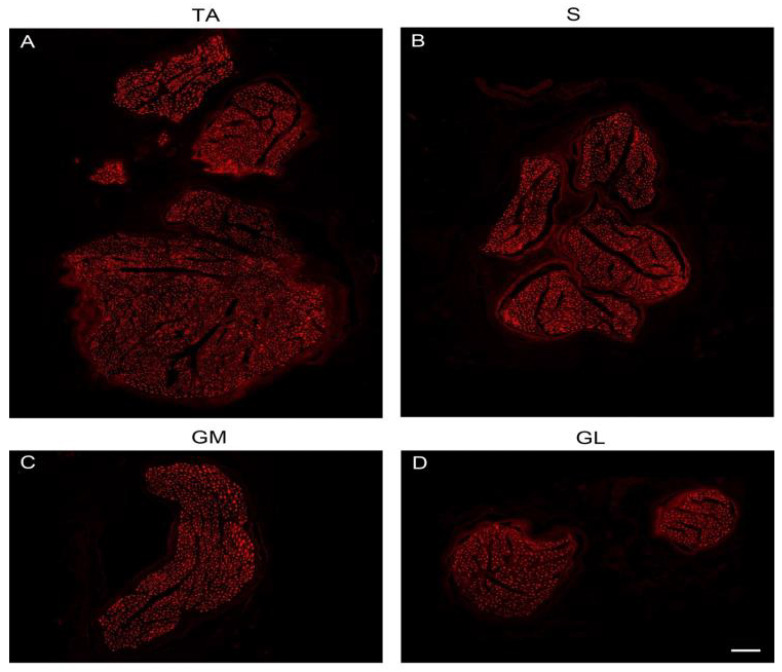
CLSM image, showing the four nerves of the lower extremity in NF staining. Scale bar, 100 µm.

**Figure 3 jpm-13-00344-f003:**
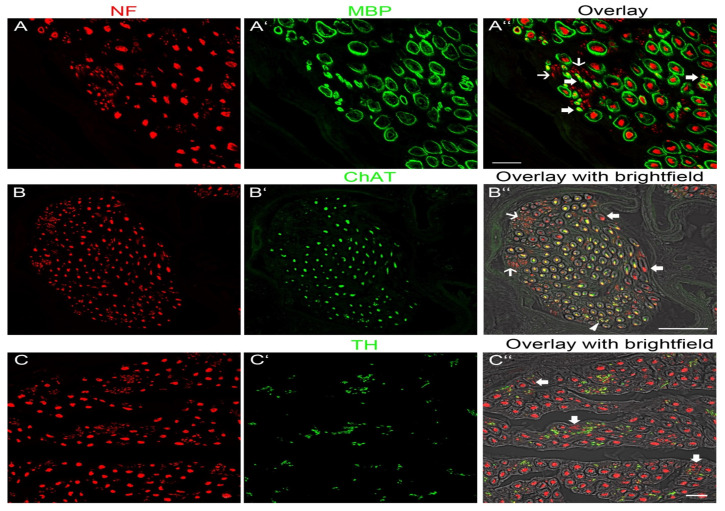
Fluorescence image showing the molecular profile of nerves innervating muscles of the lower extremity. (**A**–**A″**) Double labelling with anti-NF (red) and anti-MBP (green), a marker for myelin. (**A**) Showing axons in NF staining and (**B**) the myelin sheath in MBP staining. (**C**) Overlay visualising that large axons and thin axons (large arrows) are covered by a myelin sheath. Other thin axons (small arrows) lack a myelin sheath. Scale bar, 20 µm. (**B**–**B″**) Double labelling with anti-NF (red) and anti-ChAT (green), a marker for cholinergic axons. (**A**) Showing axons in NF and (**B**) in ChAT staining. (**C**) Overlay of fluorescence and bright field images. The bright field image shows the anatomical structure of the myelin sheath. Large myelinated axons and thinly myelinated axons (arrowhead) express ChAT. Other large myelinated axons (large arrows) lack ChAT immunoreactivitiy. Unmyelinated axons (thin arrows) lack ChAT. Scale bar, 50 µm. (**C**–**C″**) Double labelling with anti-NF (red) and anti-TH (green), a marker for sympathetic axons. (**A**) Showing axons in NF and (**B**) in TH staining. (**C**) Overlay of fluorescence and bright field images. The myelin sheath is shown in the bright field image. All myelinated axons and a few unmyelinated axons (arrows) lack TH immunoreactivity. The vast majority of unmyelinated axons express TH. Scale bar, 20 µm.

**Table 1 jpm-13-00344-t001:** Nerve specifications showing the nerve diameter, the axon count and the distance from motor entry point (MEP) to the femorotibial joint (FTJ) for the investigated muscle branches with mean and standard deviation (SD).

	Nerve Diameter	Axon Count	Distance MEP to FTJ
Recipient Nerve			
Tibialis Ant	1.97 SD 0.32	1597.14 SD 325.94	
Donor Nerve			
Lateral GCM	1.49 SD 0.37	297.5 SD 106.82	38.30 SD 10.43
Medial GCM	1.50 SD 0.32	418.5 SD 62.44	34.70 SD 10.61
Soleus	1.94 SD 0.37	1101.86 SD 135.92	66.28 SD 10.68

**Table 2 jpm-13-00344-t002:** Nerve transfer comparison showing the percentage of extremities in which the individual nerve transfers were able to reach the fibular head, the peroneal nerve bifurcation, the tibialis anterior (TA) motor entry point (MEP) as well as the mean regeneration distance for each nerve transfers to reach the targeted TA.

Donor Nerve	Reaching Fibular Head	Reaching Peroneal Bifurcation	Reaching TA MEP	Average Regeneration Distance in mm
Lateral GCM	57.69%	36.53%	0%	43.75 SD 12.1
Medial GCM	34.62%	28.86%	0%	48.31 SD 11.32
Soleus	98.08%	94.23%	13.46%	19.12 SD 11.68

## Data Availability

Data is contained within the article.

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
