# Peer review of "Distal Nerve Transfers in High Peroneal Nerve Lesions: An Anatomical Feasibility Study"

_jpm, 2023, doi:10.3390/jpm13020344_

Round 1

Reviewer 1 Report

Congratulations to the authors for the idea and the methodological quality of the study.

The English used is appropriate and does not need revision.

Abstract and background provided by the authors are comprehensive, as well as the purpose of the study is well explained and that is to evaluate and compare anatomical feasibility as well as axon count match between the tibial nerve motor branches and the tibialis anterior motor branch for a direct nerve transfer.

The only point the authors should try to make more explicit in the discussion are the future clinical developments to which the results of this study may lead.

Author Response

Point-to-point response “Distal Nerve Transfers in High Peroneal Nerve Lesions: An Anatomical Feasibility Study”

Reviewer 1

Comment:

Congratulations to the authors for the idea and the methodological quality of the study.

Reply: Thank you very much for the appreciation of our work.

Comment:

The English used is appropriate and does not need revision.

Reply: Thank you very much.

Comment:

Abstract and background provided by the authors are comprehensive, as well as the purpose of the study is well explained and that is to evaluate and compare anatomical feasibility as well as axon count match between the tibial nerve motor branches and the tibialis anterior motor branch for a direct nerve transfer.

Reply: Thank you very much for the appreciation of our work.

Comment:

The only point the authors should try to make more explicit in the discussion are the future clinical developments to which the results of this study may lead.

Reply: We agree. The discussion has been reorganized and adapted within the revised manuscript.

Reviewer 2 Report

Introduction

            The introduction presents a concise description of the peroneal nerve damage issue, its influence on gait, and the limits of current surgical restoration alternatives. It may, however, benefit from greater structure and better organization. The sentences are a little long and might be divided into shorter, more understandable phrases.

            There are some instances of repetition, notably in the usage of tendon transfers and nerve transfers, that may be minimized to enhance readability. The goal of the research, which is addressed at the conclusion but not completely defined, might also benefit from a fuller explanation in the introduction. Overall, the introduction might be better organized and more succinct.

Methods

Please address the following for your statistical analysis:

            Confidence interval: specify the confidence interval used in the analysis.

            Correction for multiple comparisons: indicate whether a correction for multiple comparisons was used, such as Bonferroni correction.

            Effect size: report the effect size measures, such as Cohen's d, to provide a more informative summary of the results.

Results

Figures are ok.

Table 1 and 2 design should be reworked. The design aspect is not friendly for readers, the font and text sizes are different.

Discussion

Please start your discussion section with the main finding of your manuscript.

Limitations of the study: Discuss any limitations of the study, such as sample size, generalizability, or any other limitations that may have impacted the results.

Conclusion

The conclusion is valid, indicating that the soleus nerve transfer is the preferred option for functional reconstruction of ankle dorsiflexion in extensive peroneal nerve lesions based on its anatomical suitability and small difference in axon counts between donor and recipient nerves

References

50% of references are 10 years old or more. Please rework and update your references.

Author Response

Point-to-point response “Distal Nerve Transfers in High Peroneal Nerve Lesions: An Anatomical Feasibility Study”

Reviewer 2

Comment:

The introduction presents a concise description of the peroneal nerve damage issue, its influence on gait, and the limits of current surgical restoration alternatives. It may, however, benefit from greater structure and better organization. The sentences are a little long and might be divided into shorter, more understandable phrases.

Reply: Thank you very much. The introduction has been revised accordingly.

Comment:

There are some instances of repetition, notably in the usage of tendon transfers and nerve transfers, that may be minimized to enhance readability. The goal of the research, which is addressed at the conclusion but not completely defined, might also benefit from a fuller explanation in the introduction. Overall, the introduction might be better organized and more succinct.

Reply: Thank you very much for your comment. The Introduction has been revised accordingly to enhance structure and readability.

Comment:

Methods

Please address the following for your statistical analysis:

            Confidence interval: specify the confidence interval used in the analysis.

            Correction for multiple comparisons: indicate whether a correction for multiple comparisons was used, such as Bonferroni correction.

            Effect size: report the effect size measures, such as Cohen's d, to provide a more informative summary of the results.

Reply: The statistical analysis was performed using standard t-tests. Multivariant comparisons were not used, therefore, no other parameters can be added.

Comment:

Results

Figures are ok.

Table 1 and 2 design should be reworked. The design aspect is not friendly for readers, the font and text sizes are different.

Reply: We agree. The font and text sizes have been adapted to the rest of the manuscript.

Comment:

Discussion

Please start your discussion section with the main finding of your manuscript.

Limitations of the study: Discuss any limitations of the study, such as sample size, generalizability, or any other limitations that may have impacted the results.

Reply: We agree. The discussion has been reorganized and adapted according to your comment. Please find changes within the revised manuscript.

Comment:

Conclusion

The conclusion is valid, indicating that the soleus nerve transfer is the preferred option for functional reconstruction of ankle dorsiflexion in extensive peroneal nerve lesions based on its anatomical suitability and small difference in axon counts between donor and recipient nerves

Reply: Thank you very much.

Comment:

References

50% of references are 10 years old or more. Please rework and update your references.

Reply: Thank you for this comment. The references have been updated including more recent articles.